# Placental Adaptation to Hypoxia: The Case of High-Altitude Pregnancies

**DOI:** 10.3390/ijerph22020214

**Published:** 2025-02-04

**Authors:** Sofia Ahrens, Dominique Singer

**Affiliations:** 1Department of Pediatric Surgery, Altona Children’s Hospital, University Medical Center Eppendorf (UKE), 20251 Hamburg, Germany; sofia.ahrens@kinderkrankenhaus.net; 2Division of Neonatology and Pediatric Critical Care Medicine, University Medical Center Eppendorf (UKE), 20251 Hamburg, Germany

**Keywords:** pregnancy, placenta, high altitude, hypobaric hypoxia, metabolic adaptation, fetal growth restriction, pre-eclampsia, highland ancestry

## Abstract

Even in the highest inhabited regions of the world, well above 2500 m altitude, women become pregnant and give birth to healthy children. The underlying adaptation to hypobaric hypoxia provides interesting insights into the physio(patho)logy of the human placenta. Although increasing altitude is regularly associated with fetal growth restriction (FGR), oxygen deficiency does not appear to be a direct cause. Rather, placental oxygen consumption is reduced to maintain the oxygen supply to the fetus. This comes at the expense of placental synthesis and transport functions, resulting in inappropriate nutrient supply. The hypoxia-inducible factor (HIF-1α), which modulates the mitochondrial electron transport chain to protect placental tissue from reactive oxygen species, plays a key role here. Reduced oxygen consumption also reflects decreased placental vascularization and perfusion, which is accompanied by an increased risk of maternal pre-eclampsia at high altitude. In native highlanders, the latter seems to be attenuated, partly due to a lower release of HIF-1α. In addition, metabolic peculiarities have been described in indigenous people that enhance glucose availability and thus reduce the extent of FGR. This review attempts to revisit the (albeit incomplete) knowledge in this area to draw the clinical reader’s attention to the crucial role of the placenta in defending the fetus against hypoxia.

## 1. Introduction

The occurrence of fetal growth restriction (FGR), which affects approximately 10 % of all pregnancies in high-income countries, has many different causes and reflects an interplay of maternal, placental, fetal, and/or genetic factors. Essentially, a mismatch between fetal nutrient requirements and placental nutrient supply leads to retarded intrauterine growth and may result in impaired fetal development and permanently altered metabolism [1,2,3].

The dynamic process of FGR must be distinguished from the more static condition of being small for gestational age (SGA): Not all SGA newborns are growth restricted (there are many other causes of low birth weights), and FGR babies do not necessarily have to be severely SGA [3]. However, perinatal mortality in full-term neonates with birth weights below the 3rd percentile is about ten times higher than in neonates with birth weights between the 25th and 75th percentiles [4]. In addition, FGR is a common cause of perinatal morbidity by increasing the risk of several postnatal complications, including perinatal asphyxia, myocardial dysfunction, and necrotizing enterocolitis [5]. This is complemented by long-term consequences such as impaired neurocognitive development and growth retardation. Affected individuals are also predisposed to a number of chronic conditions in adulthood, among them type 2 diabetes, metabolic syndrome, cardiovascular disease, osteoporosis, etc. One hypothesis that has gained the most support as an explanation for this phenomenon, also known as “Fetal Origins of Adult Disease” (FOAD), assumes intrauterine epigenetic imprinting as a result of metabolic adaptation to suboptimal maternal, placental, or fetal conditions [6,7].

Hypoxia due to incomplete hemochorial placentation is a much-discussed factor in the pathophysiology of pregnancy complications such as pre-eclampsia and FGR, but the mechanisms behind it are still poorly understood [8]. Since it is not ethically acceptable to study the effects of chronic hypoxia on human pregnancy, high altitude (HA, >2500 m above sea level) provides a natural model. In women living at HA under conditions of chronic hypobaric hypoxia during pregnancy, the effects of hypoxia on the placenta, its function and metabolism, and on the fetus can be studied in the absence of other risk factors and diseases. Notably, compared to low-altitude (LA) fetuses, HA fetuses are smaller, and pregnancies are more prone to complications such as pre-eclampsia and FGR [9,10,11,12,13,14,15]. Although oxygen supply decreases at high altitude, overall fetal oxygen uptake per kilogram body weight is not substantially reduced (details see below) [16]. Therefore, fetal tissue hypoxia is unlikely to be the major underlying cause of growth restriction. Oxygen transport from maternal circulation to the growing fetus is regulated by the placenta, which itself has a high oxygen consumption rate. Under conditions of hypobaric hypoxia, metabolic remodeling appears to occur, reducing the oxygen consumption of placental cells to maintain the oxygen supply to the fetal organism [8]. The exact mechanisms underlying this placental metabolic adjustment are not yet fully understood. Hypoxia-inducible factors (HIFs) and reactive oxygen species (ROS) are significantly involved, acting as hypoxia sensors and inducing various signaling pathways [17,18,19].

## 2. Fetal Growth at High Altitude

Fetal growth is influenced by the altitude above sea level at which the pregnancy takes place. On average, a decrease in birth weight of 100 g for every 1000 m increase in altitude can be observed, and neonates at HA are more likely to have low birth weights (<2500 g) than newborns from LA areas [9,12,14,15]. Accordingly, the fetal weight estimated by ultrasound before birth is significantly lower at HA, especially at the end of pregnancy [20]. These observations are independent of socioeconomic status and of other known risk factors for FGR, so altitude can be considered a relevant cause [9,21]. This holds true even though, as mentioned earlier, the incidence of pregnancy complications such as pre-eclampsia is also increased at HA [8,13]. The most likely cause of FGR at HA is maternal hypobaric hypoxia, as maternal conditions that result in reduced blood oxygen levels, such as cyanotic heart disease, are associated with FGR as well [20,22]. Rodent experiments confirm this assumption, in that inhalation hypoxia (F_i_O_2_ < 0.11) leads to an approximately 22 % reduced fetal weight in the exposed animals [23].

## 3. Overall Placental Properties Under Normoxia and Hypoxia

### 3.1. Placental Function at Sea Level

The placenta is an organ with various synthesis and transport functions that are associated with a very high energy demand and oxygen consumption rate. The placenta itself consumes around 40% of the oxygen available to the fetoplacental unit [16]. As the interface between the pregnant woman and the growing fetus, it also plays the role of a “gatekeeper”, regulating not only its own high oxygen requirement but also the transport of oxygen to the fetus. At the placental barrier, the exchange of oxygen and nutrients between the maternal blood in the intervillous space and the fetal capillaries takes place via diffusion, transporter molecules, and endo-/exocytotic processes [24]. Notably, at the beginning of pregnancy, the intervillous space is not yet filled with maternal blood, and the fetus is supplied by histotrophic nutrition [8]. Only when the lacunae in the syncytiotrophoblast fill with maternal blood towards the end of the first trimester does the content in the placenta rise rapidly [25]. At this point the rate of oxidative phosphorylation increases to meet the increasing energy demand of the placenta [26].

### 3.2. Placental Adaptation to High Altitude

With increasing altitude, the barometric pressure decreases continuously. For example, at an altitude of 2500 m, on the threshold to “high altitude”, it amounts to 735 hPa (550 mmHg) compared to 1013 hPa (760 mmHg) at sea level. Accordingly, the O_2_ partial pressure (pO_2_) in dry ambient air (roughly 21% of the barometric pressure) drops to 150 hPa (115 mmHg), coming from around 210 hPa (160 mmHg) at zero altitude. At an altitude of 4000–5000 m, in the Tibetan or Andean highlands, the highest regions worldwide where indigenous people live and become pregnant, the barometric pressure is around 550 hPa (410 mmHg), resulting in an ambient pO_2_ of around 110 hPa (85 mmHg). Due to an accelerated respiratory rate and to the sigmoid shape of the oxygen–hemoglobin dissociation curve, this “only” reduces arterial O_2_ saturations (SaO_2_) to 80–90% (as opposed to 95–100% at sea level) [16,20].

The reduced oxygen availability leads to hematologic changes in affected adults, especially in acclimatizing lowlanders, that partially compensate for the diminished supply. As a result of enhanced erythropoietin (EPO) synthesis, there is an elevation of the hemoglobin content, which improves the oxygen transport capacity but may also impair the flow properties of the blood (and thus the blood flow to the placenta) provided the packed cell volume (hematocrit) becomes too high [16]. In addition, the increase in 2,3-diphosphoglycerate (2,3-DPG) in the red blood cells results in a rightward shift of the oxygen–hemoglobin dissociation curve, which facilitates oxygen release but above all compensates for the leftward shift that would otherwise occur as a result of hypoxia-induced hyperventilation and hypocapnia [27].

At the tissue level, increased levels of HIF-1α were detected in the placenta of pregnant women as well as in a rat model of inhalation hypoxia [17,18]. HIF-1α, which is stabilized under hypoxic conditions, binds to the hypoxia-response elements (HRE) and acts as a transcription factor to regulate the expression of several genes [8,19]. These include genes for growth factors such as EPO (cf. above), vascular endothelial growth factor (VEGF), and selected metabolic enzymes, e.g., for anaerobic glycolysis [28]. Thus, HIF-1α contributes significantly to cell survival under conditions of hypoxia and serves as a cellular marker of hypoxia [8,29]. Under the low oxygen supply to which the placenta is exposed early in pregnancy prior to maternal vascular ingrowth, HIF-1α also plays an important role in placental vessel formation [30]. The increased vascularization occurring in early pregnancy may be regarded as an adaptation mechanism, protecting the placenta from hypoxic stress. In fact, increased vascularization occurs in placentas from pregnancies at HA compared with placentas at LA, possibly also due to increased activity of the HIF pathway at HA. Placentas from uncomplicated pregnancies at HA, which show greater vascularity, exhibit no markers of acute hypoxic stress at the end of pregnancy, such as increased HIF/DNA binding [29].

Some studies have shown increased villous vascularization, thinner villous membranes, and increased proliferation of the villous cytotrophoblast in placentas from HA compared with LA. These changes may also be interpreted as an adaptive mechanism of the placenta to hypoxia by facilitating the diffusion of oxygen from maternal blood into fetal capillaries. However, these observations are not consistent across studies [31,32,33,34]. Another study showed a smaller parenchymal surface area and slightly, although not significantly, increased thickness of the villous membrane in human placentas from high altitude (3100 m) compared to 1600 m, lowering the morphometric diffusion capacity with increasing altitude [35]. The methodology for determining the thickness of the villous membrane is very prone to error, and the variations could be due to different tissue processing and fixation. Furthermore, the sample sizes of the morphological analyses were very small, the exact altitudes between the studies differed, and no distinction was made between women with HA ancestry over several generations and women with LA ancestry [31,32,35]. In a rat model, no difference in the thickness of the diffusion barrier and the diffusion capacity between normoxic and hypoxic pregnancies could be demonstrated [36].

Figure 1 provides a summary of general adaptations to hypobaric hypoxia as well as specific hemodynamic changes and metabolic placental adaptations. Furthermore, it visualizes the influence of multi-generational HA-descent on these adaptations.

## 4. Effects of High Altitude on Placental Metabolism

### 4.1. Placental Energy Metabolism in Chronic Hypobaric Hypoxia

Aerobic mitochondrial metabolism results in a high cytosolic ATP/ADP ratio. In hypoxia or ischemia, ATP production is impaired, the ATP/ADP ratio decreases, and anaerobic glycolysis is stimulated [37]. Placental cells from HA pregnancies have a lower ATP/ADP ratio than LA placentas. Apparently, the rate of oxidative phosphorylation is lower, and anaerobic glycolysis partly accounts for energy production. At the same time, the cells show fewer signs of hypoxic/ischemic damage after labor-induced acute oxygen deprivation than in a sea-level control group. This suggests a metabolic remodeling during pregnancy at HA that adapts the placenta to hypoxic conditions by reducing ATP demand [38,39].

The observation of a markedly lower birth weight of fetuses at HA despite maintained fetal oxygen consumption also indicates metabolic remodeling [16]. The placenta appears to limit its own cellular oxygen consumption rate in order to maintain oxygen transport to the growing fetus. However, this comes at the expense of other energy-demanding processes, such as nutrient transport to the fetus [8,16]. In pregnancies at 3600 m, a decreased glucose transport to the fetus and a more than 28% reduced fetal glucose consumption rate were described, although glucose transport to the fetoplacental unit as a whole was not altered. This suggests increased placental glucose consumption, possibly due to an upregulated anaerobic glycolysis [40]. Fetal glucose deficiency may be one of the causes of the observed fetal growth restriction at HA [41]. Fetuses are not involved in gluconeogenesis and thus entirely depend on glucose transport across the placenta, which mainly occurs via glucose transporters 1–3 (GLUT1-3) [41,42]. The concentration of basal lamina-bound GLUT1 transporters is lower in HA than in LA placentas, presumably resulting in a limited capacity to transport glucose to the fetus [43,44]. Consistent with this observation, induction of oxidative stress in placental cells ex vivo results in significantly reduced expression of GLUT1 transporters and significantly lower glucose uptake [45]. However, in an experiment using BeWo chorionic carcinoma cells as a model for trophoblasts, hypoxia resulted in an upregulation of GLUT1 expression and increased transepithelial glucose transport [46]. The influence of hypoxia on placental glucose transport is not yet fully understood.

Although energy production by oxidative phosphorylation is more efficient than anaerobic glycolysis, it carries a risk of ROS formation at complexes I and III of the electron transport chain (ETC) at the inner mitochondrial membrane. Under hypoxic conditions the risk of ROS generation increases [19]. It is likely that the rate of oxygen consumption is not passively downregulated due to a lack of substrate (O_2_), but as a homeostatic adaptation to avoid excessive ROS production and subsequent cell death [47]. ROS act as important hypoxia sensors and lead to the stabilization of HIF [19,48]. HIF in turn leads to an exchange of cytochrome c oxidase (COX = complex IV of the ETC) subunits, allowing electron transfer to occur more efficiently in hypoxia. This reduces early electron transfer at complexes I and III and limits the amount of ROS produced [49]. A significant increase in HIF-1α levels and increased expression of target genes (EPO, VEGF, etc.) has also been demonstrated in cells from HA placentas [18].

Through the activation of microRNA-210 (mir-210), the HIF signaling pathway also represses the expression and correct assembly of ETC complexes I, II, and IV, thereby impairing oxidative phosphorylation in mitochondria [19]. Reduced expression of complexes I and IV, together with a sharp increase in miRNA levels, has been shown in placental fibroblasts under hypoxia. Furthermore, a 45% lower level of complex I was found in placentas from HA compared to LA [50]. Low ETC complex I activity could reduce the amount of ROS generated at complex III during hypoxia by reducing electron uptake into the ETC [51,52].

Experiments on rodents exposed to inhalation hypoxia during pregnancy support the theory of metabolic remodeling of the placenta. Hypoxic conditions with an inspiratory O_2_ fraction (F_i_O_2_) of 0.13, corresponding to an altitude of approximately 3700 m, lead to reduced mitochondrial respiration in the transport zone of the placenta with preserved oxygen transport to the fetus and unchanged or only slightly reduced fetal growth. In addition, reduced levels of ETC complex III and ATP synthase have been reported [53]. In mice, extreme inhalational hypoxia with a F_i_O_2_ of 10%, corresponding to an altitude of over 5000 m, results in FGR and in increased oxidative stress in the placental transport zone, with a concomitant reduction in amino acid transport across the placenta [53,54]. In human pregnancies at an altitude of around 2800 m, reduced amino acid transport across the placenta does not appear to be a cause of FGR [55].

### 4.2. Placental Protein Biosynthesis in Chronic Hypobaric Hypoxia

After prolonged hypoxia, cellular homeostasis is eventually disrupted, and highly energy-consuming processes are downregulated despite various adaptive mechanisms such as the HIF signaling pathway described above. These include protein biosynthesis and post-translational protein folding [56]. Unfolded and misfolded proteins are generated under oxygen deprivation and oxidative stress and accumulate in the endoplasmic reticulum (ER), resulting in impaired ER function and inducing the unfolded protein response (UPR). This is an adaptive mechanism essential for cell survival and involves the inhibition of protein biosynthesis, resulting in fewer misfolded proteins that can cause apoptosis [19,56]. Such modulation of protein synthesis has also been reported in the placenta of pregnant women at HA [57,58]. As protein biosynthesis is responsible for approximately 30% of placental oxygen consumption [26], suppression of protein synthesis in hypoxia appears to be an effective mechanism to reduce cellular oxygen consumption and maintain oxygen transport to the fetus.

Inhibition of protein biosynthesis is caused by phosphorylation of eukaryotic initiation factor 2 alpha (eIF2α) by protein kinase RNA-like endoplasmic reticulum kinase (PERK). The phosphorylated form of eIF2α is significantly increased in placental cells from HA. Apart from the maintenance of oxygen transport to the fetus, it is possible that the ER stress-induced diminution of protein synthesis also leads to FGR in HA due to the simultaneous slowing of cell proliferation [57].

Lastly, proper protein folding is also essential for mitochondrial function. Prolonged hypoxia leads to misfolded mitochondrial proteins and thus to activation of the mitochondrial UPR (UPR^mt^), which in turn leads to inhibition of mitochondrial and nuclear gene expression and apoptosis [56]. Yung et al. demonstrated activation of UPR^mt^ in conjunction with reduced oxidative phosphorylation in placentas from pre-eclamptic women [59].

## 5. Effects of High Altitude on Placental Perfusion

In early pregnancy, when the embryo is still supplied by histiotrophic nutrition, the placental pO_2_ is around 20 mmHg [8]. After hemochorial placentation towards the end of the first trimester, placental pO_2_ rises to 60 mmHg to meet the increasing oxygen and energy demands of the placenta and growing fetus [8,60]. The increase in maternal blood volume during pregnancy, increased cardiac output, decreased peripheral resistance and vasodilation of the uterine artery raise the estimated total blood flow to the uteroplacental unit to approximately 900 mL/min at the end of pregnancy, amounting to nearly 20% of maternal cardiac output [61,62,63].

In pregnant women at HA, cardiovascular changes are less pronounced. The increase in blood volume and cardiac output is smaller, and growth and remodeling of the uterine artery is also less marked, so that the blood flow to the uteroplacental unit does not increase to the same extent as in women at LA [64]. Compared to women at LA, the diameter of the uterine artery and the volumetric blood flow are smaller, and the flow velocities are higher in pregnant women at HA. The volumetric blood flow in the uterine artery is one third lower in women at 3600 m than at 1600 m altitude at 36 weeks of gestation [62].

Furthermore, at HA there is no decrease in mean arterial blood pressure until the 20th week of gestation, which is physiological in LA pregnancies. This might be a predisposition to pre-eclampsia in pregnant women at HA [11,13].

Experiments in rats yielded similar results, showing significantly lower fetal weight under hypoxia and impaired maternal vascular function ex vivo, as evidenced by an impaired vasodilatory response to methacholine [65].

Possible causes for the incomplete adaptation of maternal circulation during pregnancy under hypoxic conditions have been investigated in sheep. A study of isolated uterine arteries from pregnant and nonpregnant sheep showed that increased activity of the calcium-dependent potassium channel (BK_Ca_) in smooth muscle cells and increased Ca^2+^ spark frequency play a role in the regulation of vascular tone during pregnancy [66]. Chronic hypoxia inhibits the pregnancy-induced increase in BK_Ca_ activity and Ca^2+^ spark frequency in the ovine uterine artery, resulting in increased uterine vascular tone [67]. Reduced vasodilation via decreased BK_Ca_ activity has also been observed in myometrial arteries of pregnant women at HA compared to LA [68]. The BK_Ca_ channel appears to be involved in the pregnancy-associated decrease in myogenic tone in uterine arteries and thus plays an important role in the physiological adaptation of the uterine circulation to pregnancy [69].

Again, the underlying cause of the dysregulation of BK_Ca_ channel activity under hypoxia may be ROS. After inhibition of ROS production, the decrease in BK_Ca_ channel activity seen under hypoxia was no longer observed in ovine vessels [70]. Further analysis of uterine arteries from HA sheep also showed the role of ER stress in the increase in vascular tone under hypoxic conditions. Inhibition of the ER stress-induced PERK pathway in uterine arteries from HA sheep successfully reduced the effect of hypoxia on Ca^2+^ sparks, thereby reducing vascular myogenic tone to the level of the LA control group [67].

## 6. Intergenerational Adaptation to High Altitude

Ancestry from HA regions appears to confer protection against altitude-associated FGR. At HA, although there is a higher incidence of low birth weight in all ancestry groups, newborns from women of European descent are significantly more likely to be affected than newborns from women of HA origin. This finding is independent of other known risk factors for low birth weight [71]. Pregnancy-related increase in blood flow to the uteroplacental unit is less impaired in women of HA ancestry (from the Andes and the Tibetan highlands) than in women of LA ancestry [64,72]. One study found significantly higher uterine blood flow in women of Andean ancestry than in women of European descent living in the Andes. This resulted in a 1.6-fold increase in oxygen delivery to the uteroplacental unit at the end of pregnancy [73]. Placental transcriptome analysis showed that the expression of factors involved in placental angiogenesis and vasodilation is significantly higher in Tibetans of HA ancestry than in Han Chinese who have lived at HA for less than three generations and in LA residents. This suggests the vascular changes to be a multi-generational process rather than a rapid adaptation mechanism [58].

Possible mechanisms underlying this intergenerational adaptation may be related to the HIF signaling pathway. Endothelin-1 (ET-1) is a vasoconstrictor whose expression is regulated by HIF. People who have lived in the Andes for generations show a physiological decrease in ET-1 levels during pregnancy. In Europeans at HA, ET-1 levels are elevated, and the physiological decrease in ET-1 concentration during pregnancy is less pronounced [72]. It is known that an imbalance between the anti-angiogenic factor soluble Fms-like tyrosine kinase-1 (sFlt-1) and the pro-angiogenic placental growth factor (PLGF) is involved in the pathophysiology of pre-eclampsia. The pre-eclamptic placenta releases the anti-angiogenic factor sFlt-1, which inhibits the pro-angiogenic PLGF, resulting in a predominance of anti-angiogenesis [74,75]. Similarly, women of European descent in the Andes have higher sFlt-1 levels than women of Andean ancestry. Chronic hypoxia leads to an upregulation of sFlt-1 and downregulation of PLGF via the HIF signaling cascade. This mechanism appears to be less pronounced when there is multi-generational adaptation to high altitude, resulting in less vasoconstriction and higher blood flow in the uterine artery [76]. Correspondingly, a miRNA analysis of Tibetan and European women living at HA revealed a downregulation of HIF-1α in placentas of Tibetans who had lived at HA for several generations but not in placentas of Europeans living at HA [77].

The higher blood flow is also contributed to by greater AMP-dependent kinase (AMPK)-mediated vasodilation in women of HA ancestry [78]. Several single nucleotide polymorphisms (SNPs) have been found to be associated with uterine artery diameter and birth weight in Andean women. One of these SNPs is located near the gene for the catalytic subunit of AMPK [79]. AMPK is a hypoxia sensor that causes vasodilation of the uterine arteries in mouse studies [78,80]. This may be an important mechanism contributing to the maintenance of fetal growth to term. In pregnancies of women at HA with appropriate for gestational age (AGA) fetuses, ex vivo pharmacological activation of AMPK results in more pronounced vasodilation than in women at LA. This suggests that the myometrial arteries at HA have an increased sensitivity to AMPK, which would result in better blood flow to the uteroplacental unit at HA and thus could have a positive effect on fetal growth. However, there was no significant difference in AMPK-dependent vasodilation between FGR pregnancies and AGA pregnancies at HA in this study [81]. In a rodent model, hypoxia was shown to reduce AMPK activation in uterine arteries of pregnant mice and to reduce fetal weight by 35 %. This growth restriction was successfully counteracted by pharmacological AMPK activation, in part due to increased blood flow in the uterine artery in hypoxic pregnancy, while having no effect in normoxic mice [82]. AMPK activation merits further investigation as a potential therapy for FGR pregnancies and other hypoxic pregnancy disorders.

In addition, changes in the expression of genes involved in energy and glucose metabolism have been described in women who have lived at HA for several generations. For example, the expression of glucose transporters was found to be increased in the placentas of multi-generational Tibetans at HA compared to immigrants living at HA for less than three generations. Transcriptome analysis also suggests decreased insulin secretion and increased insulin resistance in pregnant Tibetans, which is triggered, among other things, by a significantly lower expression of placenta-specific insulin-like growth factor 2 (IGF2). Furthermore, the expression of proteins involved in glucose metabolism differs between multi-generational and immigrant HA residents. While placentas from immigrants show an up-regulation of anaerobic glycolysis, an increase in protein expression for the citric acid cycle and oxidative phosphorylation is observed in the HA ancestry group. These changes in gene expression in women with HA ancestry may support a higher glucose supply to the fetus and more efficient glucose metabolism in the placenta and may partly explain the less impaired fetal growth observed after multi-generational HA adaptation [58].

## 7. Clinical Implications and Perspectives

Studying the effects of HA on pregnancy is challenging because communities living above 2500 m in the United States are very small and widely dispersed. In the HA areas of South America and the Himalayas, many women do not give birth in a hospital, and screening, diagnostic, and treatment standards differ from those in high- and middle-income countries. In addition, not all studies differentiate between HA ancestry and recent HA immigration, making it difficult to distinguish between acute and multi-generational adaptation to hypoxia. In Europe, there are no communities living permanently at HA at all [11]. Given the geographical heterogeneity, the studies on pregnancies at HA are only comparable to a limited extent, both with each other (in terms of exact altitude, environmental conditions, dietary habits, etc.) and with “reference” pregnancies in high- and middle-income countries. The difficult circumstances of studying pregnancies at HA also explain the sometimes very small groups investigated, for example, in the uterine artery blood flow studies [62,77,83]. Moreover, examination methods such as (Doppler) sonography, used to analyze the blood flow in the uterine artery, are not readily available everywhere and depend on the investigator’s skills. Nevertheless, there is a high degree of consistency in the results of the numerous studies included. Reduced birth weight, increased incidence of SGA and FGR, and altered circulatory adaptation are uniformly described in multiple studies [9,10,11,13,14,15,62,64]. Animal studies of pregnancy under controlled inhalation hypoxia also support these findings [17,23,54]. It is therefore unlikely that the above confounding factors have a decisive influence on the observed differences between pregnancies at HA and LA.

HA pregnancies which are associated with (hypobaric) hypoxia and an increased risk of pre-eclampsia and FGR, may provide a model for pregnancy disorders that, although occurring at LA, are often also associated with reduced blood flow to the placenta and subsequent hypoxic stress [8]. Pre-eclampsia, a hypertensive pregnancy disorder characterized by arterial hypertension and proteinuria, affects approximately 3–5% of pregnancies and is a leading cause of maternal, fetal, and neonatal mortality [84]. Much of the pathophysiology of preeclampsia is still unknown. The most common theory assumes defective remodeling of the endometrial spiral arteries, which in normal pregnancies dilate from approximately 10 weeks of gestation due to trophoblast cell ingrowth, resulting in increased blood flow to the placenta [85]. Inadequate placental blood supply may lead to a complex process of ischemia and reperfusion, exposing the placenta to alternating periods of oxygen deprivation and reoxygenation. This so-called “hypoxia–reoxygenation” (HR) results in hypoxic stress and is associated with increased generation of ROS, endothelial dysfunction, and apoptosis of trophoblasts [85]. It is likely that here, similar molecular mechanisms and signaling cascades are involved as in high-altitude pregnancies, which may also be of interest for therapeutic options. For example, Nuzzo et al. investigated the potential of mitochondria-targeted antioxidants to protect against mitochondrial oxidative stress in hypoxic pregnancies. In animal experiments, these increased placental blood exchange area and blood volume in placental lacunae and prevented the development of mitochondrial stress and activation of the UPR [36]. Nevertheless, it should be kept in mind that the physiological processes at HA, reflecting the adaptability of the healthy placenta to hypobaric hypoxia, may differ from the pathophysiological processes occurring in pre-eclampsia, which involve an undersupply due to primarily impaired placental function.

From a broader perspective, there are three different modes of adaptation to hypoxia that can be applied to the placental response to HA. Firstly, the oxygen supply can be increased by increasing the blood flow. This occurs in the placenta, as described above, as a first step of adaptation to HA through increased capillary density of the villi and increased secretion of angiogenic and hematopoietic factors. If this so-called oxyregulatory response fails to maintain the balance between oxygen demand and supply, energy metabolism will *passively* decrease, and energy production will switch to anaerobic glycolysis. However, this is usually insufficient to maintain the high metabolic rate in mammalian tissues, leading to functional impairment and, if prolonged, structural damage. However, there is a third way of adaptation to hypoxia, i.e., the *active* reduction of oxygen demand in response to decreasing oxygen supply. Although this so-called oxyconforming response, also known as hypoxic hypometabolism, is also accompanied by a temporary loss of function, it delays the onset of permanent damage. Moreover, it is a prerequisite for anaerobic glycolysis to eventually make a reasonable contribution to energy turnover [16,86], especially if it is additionally fed by an increased substrate (glucose) supply. Although this mode of adaptation was long thought to be restricted to invertebrates and lower vertebrates, it appears to be conserved in the mammalian placenta, explaining its spectacular ability to protect the fetus from severely hypoxic environmental conditions [86,87,88].

Recent evidence, partly derived from MRI studies, suggests that some kind of hypoxic hypometabolism even occurs in the human fetus, in that fetal oxygen consumption may also vary depending on oxygen supply [8,16,89]. This could be due to variable intrauterine growth rates or to hemodynamic adaptations redirecting blood flow to the central organs and thus leading to a more economic use of oxygen (“brain sparing effect”) [86,90]. However, it may also reflect true oxyconforming adaptations at the tissue level, as described in the brain of the fetal llama, one of the various mammalian species besides humans that live and successfully reproduce in the high mountain areas of the world [91]. Similar effects have also been observed immediately after birth in various mammalian species, where they appear to form a basis for the temporarily increased neonatal hypoxia tolerance [87,92,93,94]. Overall, these findings mean that the uterus, the placenta, and the fetus itself work together like a fine-tuned machine, buffering adverse ambient conditions to the greatest possible extent in favor of the growing offspring.

However, the impressive adaptation to aggravated deficiencies comes at a price. To survive in an oxygen- and/or nutrient-depleted environment in utero, whether due to HA or pre-eclampsia, a change in gene expression seems to occur that persists beyond the perinatal period and increases the risk of developing various diseases in adulthood. This type of epigenetic imprinting predisposes FGR children to metabolic syndrome, cardiovascular disease, kidney and liver disease, cancer, and many other conditions [7,95].

Just as intrauterine undersupply can create epigenetic imprinting, the genetic background appears to play a significant role in successful long-term adaptation to HA. It has long been known, for example, that Sherpas in the Tibetan highlands exhibit an amazing physical capacity and a remarkably low risk of mountain sickness, apparently due to metabolic adaptations in muscle tissue and a lower tendency to pulmonary vasoconstriction/hypertension [96,97,98]. This is exactly in line with what has been found in the placenta of pregnant women of HA ancestry, suggesting that placental function under hypobaric hypoxia is only part of a more general interplay of adaptations that make their carriers “born for the HA environment”.

Of note, metabolic changes like those observed in the placenta at HA also occur in other organs under pathological conditions, e.g., in the kidney in septic shock. When severe hypoxia occurs in sepsis due to impaired organ perfusion, ATP generation via oxidative phosphorylation is impaired, and apoptosis signaling pathways are triggered, which can result in organ damage. However, this is not initially the case in sepsis-related organ failure, as the cells can apparently adapt to the decreasing energy supply. Renal function is often impaired, although renal blood flow is still maintained or even increased [99]. One possible explanation for this is a reduction in cellular energy demand due to a decrease in the metabolic activity of the cells. This creates a new steady state that does not allow normal organ function but prevents a critical drop in ATP levels that would lead to cell death. A possible signal for the cells to switch metabolism could be ROS, which circulate in large quantities in sepsis. This assumption is supported by the minimal histological organ damage and the rapid recovery of organ function after sepsis-induced acute kidney injury. Furthermore, incubation of cells in plasma from septic patients leads to an increase in oxidative stress and a decrease in mitochondrial respiratory chain activity [100]. This calls into question the long-held assumption that sepsis-related acute kidney injury is mainly due to decreased renal perfusion. It is possible that the metabolic switch triggered by hypoxia and ROS is not, or not solely, part of the pathomechanism of organ injury, but to a certain extent also includes a self-protective response [87].

Unfortunately, the influence of (hypobaric) hypoxia on pregnancy has not been sufficiently explored, and only a few recent studies exist. Many of the described molecular mechanisms of adaptation and maladaptation of the placenta, its blood flow, and metabolism have only been investigated fragmentarily and do not yet provide a complete picture. Further studies on the transferability of the findings from HA pregnancies to pregnancy complications such as pre-eclampsia will be of considerable interest. HA pregnancies do not only reflect the striking adaptive capacity of the utero–placento–fetal unit but may also promote the understanding of hypoxia-associated pregnancy disorders and the development of novel therapeutic approaches.

## Figures and Tables

**Figure 1 ijerph-22-00214-f001:**
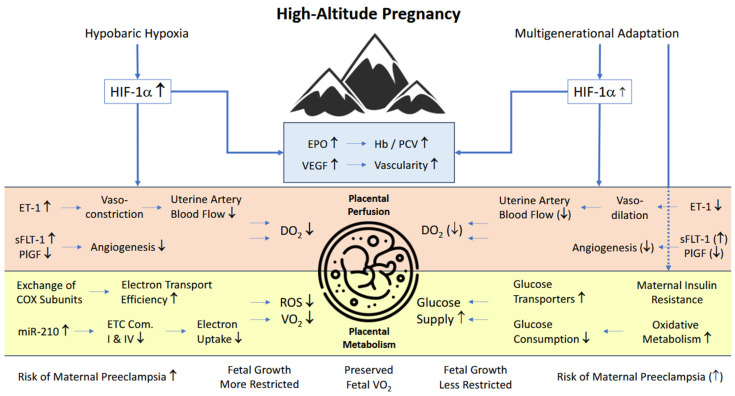
Effects of hypobaric hypoxia on placental perfusion and metabolism, a synoptic view with special respect to multi-generational adaptation (right-hand part). Reduced oxygen delivery is counteracted by both an increase in Hb and vascularity and a decrease in oxidative metabolism, reflected in fetal growth restriction. Multi-generational adaptation affects both placental perfusion and metabolism. With less reduced uterine artery flow, the risk of preeclampsia is less elevated. Due to improved glucose supply, fetal growth is less restricted. Attenuated effects are indicated by bracketed arrows.

## Data Availability

Not applicable.

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
