# Peer review of "Placental Adaptation to Hypoxia: The Case of High-Altitude Pregnancies"

_ijerph, 2025, doi:10.3390/ijerph22020214_

Round 1
Reviewer 1 Report
Comments and Suggestions for Authors
The paper provides an extensive and well-structured review of the current state of research in placental adaptation to hypoxia. The authors have effectively selected and covered a wide range of relevant literature, ensuring that the scope of the review is both clear and aligned with the subject matter. The inclusion of significant studies reflects a solid understanding of the field. The integration of the literature is well-executed, offering readers a clear view of the key research and the issues involved. This review will likely help readers gain a better understanding of the current research landscape and identify areas for future exploration. I have only a few comments, which are as follows:
Overall placental properties under normoxia and hypoxia section
1. In the last paragraph of the subsection “Placental adaptation to high altitude”, you mention: “Some studies have shown increased villous vascularization, thinner villous membranes, and increased proliferation of the villous cytotrophoblast in placentas from HA compared with LA. These changes may also be interpreted as an adaptive mechanism of the placenta to hypoxia by facilitating the diffusion of O2 from maternal blood into fetal
capillaries. However, these observations are not consistent across studies.” However, you did not provide details on how these observations are inconsistent. Briefly summarizing their results would help readers better understand the inconsistency you pointed out.
Figure
2. Figure 1 is very clear, concise, and provides sufficient details. I would recommend adding a sentence somewhere in the main text to lead readers to view Figure 1 earlier, rather than at the end of the paper, helping them visualize the mechanism while reading.
Author Response
Reviewer I
Comment: The paper provides an extensive and well-structured review of the current state of research in placental adaptation to hypoxia. The authors have effectively selected and covered a wide range of relevant literature, ensuring that the scope of the review is both clear and aligned with the subject matter. The inclusion of significant studies reflects a solid understanding of the field. The integration of the literature is well-executed, offering readers a clear view of the key research and the issues involved. This review will likely help readers gain a better understanding of the current research landscape and identify areas for future exploration. I have only a few comments, which are as follows:
Response: Thank you for your kind words.
Comment
Overall placental properties under normoxia and hypoxia section
In the last paragraph of the subsection “Placental adaptation to high altitude”, you mention: “Some studies have shown increased villous vascularization, thinner villous membranes, and increased proliferation of the villous cytotrophoblast in placentas from HA compared with LA. These changes may also be interpreted as an adaptive mechanism of the placenta to hypoxia by facilitating the diffusion of O2 from maternal blood into fetal capillaries. However, these observations are not consistent across studies.” However, you did not provide details on how these observations are inconsistent. Briefly summarizing their results would help readers better understand the inconsistency you pointed out.
Response: Thank you for this important comment. We have now added some examples to show the variety of data on morphological changes in the placenta under hypoxic conditions, including two additional references (in return for omitting two others, one German and one rather outdated). By the way, the references have been adapted to the journal style.
Comment
Figure
Figure 1 is very clear, concise, and provides sufficient details. I would recommend adding a sentence somewhere in the main text to lead readers to view Figure 1 earlier, rather than at the end of the paper, helping them visualize the mechanism while reading.
Response: Thank you for your comment. We have now added the sentence “Figure 1 provides a summary of general adaptations to hypobaric hypoxia as well as specific hemodynamic changes and metabolic placental adaptations. Furthermore it visualizes the influence of multi-generational HA-descent on these adaptations.” at the end of the subsection “Overall placental properties under normoxia and hypoxia” to alert readers to the illustration at an early stage.
Reviewer 2 Report
Comments and Suggestions for Authors
This is an extremely well thought out and organized review paper. The description of the physiological mechanisms by which hypoxia during HA causes changes to the placenta and subsequently how the placenta adapts to those changes in order to protect the fetus was well constructed and highly detailed. Particularly, the analysis of the metabolic remodeling due to the decrease in ATP/ADP ratio and placental glucose regulation in HA was well characterized. The quality of English used is very good; the review is very easy to follow and understand. The included figure (Figure 1) effectively and comprehensively conveys the subject matter of the review and aids in overall understanding. Though, if at all possible, it would be beneficial for it to be edited for uniformity and clarity. For example, the arrow from maternal insulin resistance to an increase in glucose transporters is incomplete, maternal insulin resistance is a different color to the rest of the text, etc.
The only minor edits I would suggest would be the authors checking for grammar issues such as sentace structure and formatting issues like double spacing. Additionally, they can re-edit the included figure as it does appear to have a number of uniformity and clarity issues.
Author Response
Reviewer II
Comment
This is an extremely well thought out and organized review paper. The description of the physiological mechanisms by which hypoxia during HA causes changes to the placenta and subsequently how the placenta adapts to those changes in order to protect the fetus was well constructed and highly detailed. Particularly, the analysis of the metabolic remodeling due to the decrease in ATP/ADP ratio and placental glucose regulation in HA was well characterized. The quality of English used is very good; the review is very easy to follow and understand. The included figure (Figure 1) effectively and comprehensively conveys the subject matter of the review and aids in overall understanding.
Response: Thank you for your kind words.
Comment
Though, if at all possible, it would be beneficial for it to be edited for uniformity and clarity. For example, the arrow from maternal insulin resistance to an increase in glucose transporters is incomplete, maternal insulin resistance is a different color to the rest of the text, etc.
The only minor edits I would suggest would be the authors checking for grammar issues such as sentace structure and formatting issues like double spacing. Additionally, they can re-edit the included figure as it does appear to have a number of uniformity and clarity issues.
Response: Thank you for your feedback, especially for pointing us to the fact that some of the stylistic subtleties of the illustration were not as self-explanatory as we had been hoping. As a result, we have not only made the figure more consistent, but have also revised the legend to make it (hopefully) even more understandable. In addition, we have gone through the text again and fixed the issues you raised in formatting and sentence structure; the details can be found in the file provided in the “track changes” mode.
Reviewer 3 Report
Comments and Suggestions for Authors
The review by Ahrens & Singer summarizes the current data on the hypoxia effects on the placenta in regard of high-altitude pregnancies. Still, there are several issues to be addressed.
I suggest Abstract should clearly identify the aim of the review and stress what makes this review different from other reviews on the topic. This notion seems important because the Reference list includes a large portion of other review articles.
There are several unconventional terms, use of which cannot be explained by language issues because the overall text is written in excellent English. Here follow the examples
· I suggest replacing ‘O2’ with ‘oxygen’ at least where the narration does not directly deal with molecular oxygen. That is, the phrase ‘reactive O2 species’ (Lane 11 in Abstract) looks unconventional because O2 is a chemical formula for molecular oxygen, but in ROS, oxygen atoms are present in another chemical state.
· The hypoxia-response elements (HRE) are not genes as the text reads on Page 3. HRE are cis-acting DNA sequences to recruit transcription factors.
· Another unconventional phrase is ‘…the translation of ETC complexes I, II, and IV’ (Page 4). Each of these complexes consists of multiple subunits, therefore ‘translation’ cannot be attributed to whole complexes.
· ‘ETS complex III’ (Page 4) should be replaced with ‘ETC complex III’. Do the authors mean ‘electron transport system’? It is unconventional.
In my case, finding a Discussion section in a review is confusing because a review by its nature is discussion itself. Maybe Conclusions or Perspectives? Throughout the text, there are very scarce if any critical data analysis except some in ‘Discussion’. If the authors chose to present accumulated data and their views separately, I suggest to denote the review structure in Abstract in order to avoid possible confusion of a reader.
Additionally, findings described in the review are summarized in Fig.1, however in the text, there is no reference to Fig. 1. Description of this rather complicated figure is restricted to a concise figure legend.
Collectively, I am very sorry to announce that usage of unconventional terms accompanied by my failure to identify any papers by the first author in the Reference list, and modest analysis of data makes me, as a regular reader, less trustful to the Review and makes me suspect some machine-based contribution. I apologize for the latter concern and wish the authors would resolve the raised issues.
Author Response
Reviewer III
Comment
The review by Ahrens & Singer summarizes the current data on the hypoxia effects on the placenta in regard of high-altitude pregnancies. Still, there are several issues to be addressed. I suggest Abstract should clearly identify the aim of the review and stress what makes this review different from other reviews on the topic. This notion seems important because the Reference list includes a large portion of other review articles.
Response: Thank you for pointing this out. We had to shorten the abstract a little anyway, so we took the opportunity to change the last sentence accordingly.
Comment
There are several unconventional terms, use of which cannot be explained by language issues because the overall text is written in excellent English. Here follow the examples
I suggest replacing ‘O2’ with ‘oxygen’ at least where the narration does not directly deal with molecular oxygen. That is, the phrase ‘reactive O2 species’ (Lane 11 in Abstract) looks unconventional because O2 is a chemical formula for molecular oxygen, but in ROS, oxygen atoms are present in another chemical state.
Response: Thank you for the valid point. In an effort to shorten the text, we had replaced “oxygen” with “O2”, which led to “O2” appearing in places where it would have been better to use “oxygen”. We have now replaced “O2” with “oxygen” again for reasons of correctness and consistency.
The hypoxia-response elements (HRE) are not genes as the text reads on Page 3. HRE are cis-acting DNA sequences to recruit transcription factors.
Response: Thank you for the important note. The sentence has been reworded accordingly (see the “Track Changes” file).
Another unconventional phrase is ‘…the translation of ETC complexes I, II, and IV’ (Page 4). Each of these complexes consists of multiple subunits, therefore ‘translation’ cannot be attributed to whole complexes.
Response: Thank you for this well-founded comment. The sentence has been reworded accordingly (see the “Track Changes” file).
ETS complex III’ (Page 4) should be replaced with ‘ETC complex III’. Do the authors mean ‘electron transport system’? It is unconventional.
Response: Sorry, spelling mistake, corrected. Thank you for your attention.
Comment
In my case, finding a Discussion section in a review is confusing because a review by its nature is discussion itself. Maybe Conclusions or Perspectives? Throughout the text, there are very scarce if any critical data analysis except some in ‘Discussion’. If the authors chose to present accumulated data and their views separately, I suggest to denote the review structure in Abstract in order to avoid possible confusion of a reader.
Response: You are, of course, right (not a “discussion” in the conventional sense), which is why we have now exchanged “discussion” for “clinical implications and perspectives”.
Comment
Additionally, findings described in the review are summarized in Fig.1, however in the text, there is no reference to Fig. 1. Description of this rather complicated figure is restricted to a concise figure legend.
Response: Thank you for your comment. We have now added a reference to Fig. 1 at the end of the subsection “Overall placental properties under normoxia and hypoxia” to draw the readers’ attention to the figure at an early stage. In addition, we have also revised the legend to (hopefully) make the figure even better understandable.
Comment
Collectively, I am very sorry to announce that usage of unconventional terms accompanied by my failure to identify any papers by the first author in the Reference list, and modest analysis of data makes me, as a regular reader, less trustful to the Review and makes me suspect some machine-based contribution. I apologize for the latter concern and wish the authors would resolve the raised issues.
Response: We apologize for the uncommon words or phrases we unintentionally used as we are no native English speakers. We declare that no artificial intelligence or similar was used to write the review. Additional comment from DS: Ms. SA is an exceptionally talented young collaborator who, as a student, wrote a German seminar paper on the adaptation of the placenta to high altitude. I therefore asked her to rewrite this paper under my supervision as a full publication in English language. The reason she is not listed as an author in other references is that this is simply (and unbelievably!) her very first scientific paper. This might also be the reason for some unconventional wording and has nothing to do with artificial intelligence, but in this case with natural intelligence. I hope you will accept and appreciate this.
Round 2
Reviewer 3 Report
Comments and Suggestions for Authors
In the revised version, the authors introduced proper amendments according to the issues raised by the first-round review. I have no more comments